# PNANet: Probabilistic Two-Stage Detector Using Pyramid Non-Local Attention

**DOI:** 10.3390/s23104938

**Published:** 2023-05-21

**Authors:** Di Zhang, Weimin Zhang, Fangxing Li, Kaiwen Liang, Yuhang Yang

**Affiliations:** 1School of Mechatronical Engineering, Beijing Institute of Technology, Beijing 100081, China; 3120210157@bit.edu.cn (D.Z.); wonk2000@bit.edu.cn (F.L.); liangkaiwen@bit.edu.cn (K.L.); yuhang0702@gmail.com (Y.Y.); 2Key Laboratory of Biomimetic Robots and Systems, Ministry of Education, Beijing Institute of Technology, Beijing 100081, China; 3Beijing Advanced Innovation Center for Intelligent Robots and Systems, Beijing 100081, China

**Keywords:** probabilistic two-stage detector, pyramid non-local attention, robust proposal generator, object detection

## Abstract

Object detection algorithms require compact structures, reasonable probability interpretability, and strong detection ability for small targets. However, mainstream second-order object detectors lack reasonable probability interpretability, have structural redundancy, and cannot fully utilize information from each branch of the first stage. Non-local attention can improve sensitivity to small targets, but most of them are limited to a single scale. To address these issues, we propose PNANet, a two-stage object detector with a probability interpretable framework. We propose a robust proposal generator as the first stage of the network and use cascade RCNN as the second stage. We also propose a pyramid non-local attention module that breaks the scale constraint and improves overall performance, especially in small target detection. Our algorithm can be used for instance segmentation after adding a simple segmentation head. Testing on COCO and Pascal VOC datasets as well as practical applications demonstrated good results in both object detection and instance segmentation tasks.

## 1. Introduction

Object detection plays a crucial role in robotics. For instance, in the context of household serving robots, achieving an accurate and reliable grasp of objects requires the robot to be able to acquire the precise locations of objects [1]. Object detection can also be used in the field of industrial robots to assist robots in tasks such as item sorting, component assembly, and work area confirmation [2]. Over the years, numerous studies have focused on creating precise and speedy detectors to cater to the needs of robots and other domains. Enhancing the interpretability and accuracy of detectors by optimizing their structure, as well as improving their performance in detecting and segmenting small objects, remain critical and challenging issues that current algorithms are striving to solve and overcome.

Object detectors generally fall under two categories, namely, two-stage object detectors and one-stage object detectors. Standard two-stage object detectors locate all possible object positions by maximizing the recall rate in the first stage but identify objects within these positions based on their likelihood scores. The optimization objectives in the two stages are distinct, which results in a lack of probabilistic interpretation and structural redundancy in standard second-stage object detectors. One-stage detectors maximize the likelihood of annotated ground-truth objects during the training stage and rely on the likelihood scores as the basis for inference. They are a probabilistically sound framework but the problem of insufficient accuracy may arise due to the impact of imbalanced positive and negative samples. CenterNet2 [3] modified the structure of the standard two-stage detectors and developed a probabilistic two-stage detection framework by maximizing a lower limit for a combined probabilistic goal across both stages. However, there are still limitations in the proposed approach in CenterNet2. For example, the localization quality score and classification score are trained separately but are utilized during inference in the first stage; this inconsistency between training and prediction leads to insufficient interpretability and low efficiency of the model. The positive sample selection approach during the training phase is relatively simple, which can result in lower-quality proposal boxes provided by the first stage, ultimately affecting the performance of the model. Generalized focal loss (GFL) [4] and adaptive training sample selection (ATSS) [5] have addressed the aforementioned issues to some extent, but they still lack strong prior guidance during training and inference, which can result in the incomplete probabilistic interpretation of the model and relatively weak stability. In summary, a detector with complete probabilistic interpretation and compact structure is the current focus of research.

The precise detection of small targets is another important issue in the field of object detection. There have been numerous works aiming to solve these problems. Feature pyramid network (FPN) [6] is the pioneer of those works; it has been widely adopted due to its capability of improving the detection accuracy for small targets and enhancing adaptability to multi-scale objects. Path aggregation net (PANet) [7], NAS-FPN [8] and other studies [9,10,11] have furthered the progress of network architectures for cross-scale feature integration. How to effectively integrate features from different layers, explore the correlations between them, and preserve and restore the details of the images is the current research focus. Attention mechanisms have emerged as another means of mining and preserving detailed information in recent years [12]. They enhance the accuracy and efficiency of a neural network by weighting the input data and highlighting the important parts. Some recent research [13] implies that there are interdependent relationships among pixels, and these dependencies are not limited to adjacent pixels. Pixels that are far apart from each other also have interdependencies. For example, in an image of a cat, the shape of the tail may depend on the position of the ears, even if they are far apart. Another example could be the relationship between the background color and the color of an object in the foreground, which can impact the overall visual coherence of the image. Leveraging this type of long-range dependency has the potential to enhance performance. However, such methods require a significant amount of computational resources, and methods that exclusively rely on convolutions demonstrate limited capability in capturing long-range dependencies. Only a minority of approaches have endeavored to exploit features across varying levels to capture long-range dependencies, and most of them still struggle to adequately address the computational burden involved [14]. Therefore, balancing the demands of enhancing algorithms’ ability to integrate and extract detailed features, improving their capacity for detecting and segmenting small targets, and ensuring computational efficiency is a major challenge in current research.

In this paper, we proposed a probabilistic two-stage detector that has a reasonable probability interpretation and a compact structure, enabling accurate object detection. Upon the integration of a simple segmentation header, our detector further achieves precision instance segmentation. Notably, our detector exhibits notable control over details, thereby demonstrating exceptional performance in detecting objects.

Specifically, we first introduced a robust single-stage object detector as a replacement for the region proposal network (RPN) in standard two-stage detectors. We trained both stages simultaneously to maximize the likelihood of ground-truth objects, which is then used as the detection score during inference. Secondly, we enhanced the method of ground-truth matching and improved the first-stage proposal generator by coupling the classification branch with the box generation branch and incorporating a better prior for the box regression branch. This resulted in a more stable first stage and a more comprehensive probability interpretation. Thirdly, we proposed an effective pyramid non-local attention (PNA) module, we incorporate the non-local attention mechanism into FPN to capture non-local dependency across multiple levels and embed a pyramid sampling module into every non-local block, which significantly reduces computational overhead while preserving semantic features. Finally, we made minor modifications to BiFPN, resulting in improved accuracy. Our main contributions can be summarized as follows:

1. We built a probabilistic two-stage detector that achieves higher accuracy with a more reasonable probability interpretation.

2. We proposed a strong proposal generator by coupling different branches and providing a prior for box regression. This makes the first stage more stable and interpretable, thus improving the overall accuracy of the network with almost no cost.

3. We proposed a pyramid non-local attention(PNA) module, which enhances the network’s ability to extract detailed features, ultimately significantly improving its detection capabilities for objects, especially for small objects.

The rest of this paper is outlined as follows. In Section 2, we summarize relevant work. In Section 3, we elaborate on the structure of the object detector, including the design of the strong proposal generator and PNA module in detail. Section 4 shows the experimental results. Finally, we present certain conclusions and outline our prospective research endeavors.

## 2. Related Works

Object detectors:Two-stage detectors, such as regions with CNN feature (RCNN) series [15,16,17], employed an RPN for generating imprecise object proposals, followed by using a specialized head for each region to refine and classify them. Cascade RCNN [18] improved localization accuracy by repeating the detection head of Faster-RCNN multiple times, each time utilizing different threshold values. To further improve the feature flow between stages in Cascade RCNN, hybrid task cascade (HTC) [19] incorporated extra annotations for both instance and semantic segmentation. Mask RCNN [20] is an extension of Faster RCNN that includes an instance segmentation branch for generating precise masks of the objects. Task-aware spatial disentanglement (TSD) [21] separated the localization and classification branches for each region of interest (ROI). Libra RCNN [22] and gradient harmonizing mechanism (GHM RCNN) [23] proposed new loss functions, optimizing the performance of detectors across different scales, difficulty levels, and object categories. Ammar et al. [24] enhanced models’ accuracy by expoiting the temporally redundant information. Two-stage object detectors still achieve high accuracy nowadays, but their efficiency is low due to weak proposal generators that generate numerous but low-quality proposals [3]. In addition, the two-stage optimization objectives differ, and there are discrepancies between training and evaluation metrics, resulting in a significant degradation of the overall detector performance.

One-stage detectors, such as the you-only-look-once (YOLO) series [25,26,27,28,29,30], simultaneously forecast both the object’s location and output class. The YOLO series of detectors utilize the grid-based approach to predict class and bounding box regression. Betti and Tucci [31] optimized the parameters of YOLO, further reducing the computational cost. Fully convolutional one-stage object detector (FCOS) [32] and CenterNet [33] abandoned the use of numerous anchors per pixel and determine foreground/background by location. ATSS [5] and probabilistic anchor assignment (PAA) [34], which are derived from FCOS, revised the definition of foreground and background to make the allocation of positive and negative samples more reasonable. GFL [4] provided a weighted representation of category truth values and takes into account the uncertainty of bounding boxes under occlusion, which further increased the interpretability of the algorithm. CornerNet [35] detected the two diagonals of an object; ExtremeNet [36] detected four extreme points of an object and used an additional center point to group them. RepPoint [37] and Dense RepPoint [38] utilized a set of points to represent the boundaries of bounding boxes, and the features of these points were employed to classify the objects. This type of detector often has comprehensive probability explanations, but they still lack accuracy. For example, under the same training conditions, Faster RCNN outperforms single shot multiBox detector (SSD) by five points on the COCO dataset and Cascade RCNN outperforms RetinaNet by 3.7 points on the COCO dataset.

In recent years, there has been a high level of research interest in visual transformers. The visual transformers (ViT) [39] algorithm attempted to directly apply the standard Transformer structure to images by splitting the entire image into small image blocks, and then using the linear embedding sequence of these blocks as the input to the Transformer network for training. Data-efficient image transformers (DeiT) [40] improved the training strategy based on ViT, reducing the computational resources required during training. Detection transformer (DETR) [41] replaced traditional object detection methods such as RPN and ROI Pooling with Transformer networks, greatly simplifying the object detection process. Deformable DETR [42] added deformable convolution modules to DETR to adapt to changes in object shape and size. Sparse RCNN [43] used sparse attention mechanisms to only compute regions relevant to the object. DETR with improved denoising anchor boxes (DINO) [44] algorithm achieved feature extraction and classification by using a self-attention mechanism. The use of attention mechanism and transformer can greatly improve the performance of the algorithm, but it also requires a large amount of computing power. Balancing the accuracy and computational cost is the current focus of research.

Feature pyramid: The utilization of a feature pyramid can enhance the network’s resolution, improving the detection accuracy of small objects. One of the primary challenges is to efficiently encode and handle features across multiple scales. FPN [8] proposed a top-down feature fusion structure, which greatly improves the performance of the network. Following the idea of FPN, PAN [7] added a feature aggregation path from bottom to top based on FPN, allowing for more comprehensive feature fusion. Han et al. [45] combined super-resolution with YOLOv5 to achieve improved accuracy in safety helmet detection. Scale-transferrable detection network (STDN) [46] introduced a transfer module to the network for extracting features from different scales and SNIPER [47] added a weakly supervised mechanism on top of FPN; the addition of an attention mechanism enables the network to achieve higher accuracy under the same time complexity. M2det [48] used a U-shape module to process feature fusion of different scales. Gated feedback refinement network (G-FRNet) [49] introduced gate units to regulate the flow of information between features. NAS-FPN and NAS-FPN+ [50] can automatically search for the optimal network structure, but require thousands of GPU hours during the training phase. BiFPN [51] utilized bidirectional feature fusion to merge feature maps of different levels, which balances algorithm speed and performance better than NAS-FPN. The ultimate goal of all the above methods is to fully explore valuable information from different levels and fuse them more comprehensively.

Attention mechanism: Attention mechanism plays an important role in human visual perception. In 2017, Vaswani et al. [12] introduced this mechanism into the field of machine learning, and since then, it has been widely applied. Wang et al. [52] proposed a Network that incorporates an encoder and a decoder to implement attention mechanisms, while Hu et al. [53] leveraged a Squeeze-and-Excitation module to exploit the inter-channel relationship of the Network. These approaches yielded a notable improvement in the accuracy of the algorithm. Similarly, Chen et al. [54] utilized weight matrixes to amplify salient features and suppress irrelevant ones, resulting in increased accuracy and sensitivity to small targets. Meanwhile, convolutional block attention module (CBMA) [55] and DANet [56] combined spatial and channel attention. Despite their effectiveness in enhancing the algorithm’s performance, all these methods were limited to a single scale.

Recent studies have also focused on how to make sufficient use of long-range dependencies. Wang et al. [13] proposed a non-local attention mechanism module in 2018, which was initially used for image denoising and later applied to image super-resolution in 2020 [57]. Zhang et al. [58] introduced a self-attention generative adversarial network, which uses non-local attention mechanisms to improve the details and texture of the image. Residual non-local attention networks (RNAN) [59] adopted a kind of network structure based on residual blocks and introduces non-local attention modules to capture long-range dependencies in the image. It has achieved excellent performance in multiple image restoration tasks. Zhou et al. [60] used non-local attention mechanisms for multi-organ semantic segmentation in 2019, greatly improving the accuracy and robustness of image segmentation. Many studies have shown that non-local attention mechanisms can enhance the network’s ability to extract details, but there is still relatively little research on applying non-local attention mechanisms to object detection and segmentation. Even fewer studies consider the comprehensive use of non-local attention mechanisms and multi-scale information.

## 3. Materials and Methods

The architecture of our proposed object detector is shown in Figure 1. The input image is processed by a backbone network to extract features and then downsampled to generate five features of different scales. These features are fused through a repeated feature pyramid structure, which is based on the structure proposed in EfficentDet [51] but has been improved to further consider the importance of different channels. The aforementioned features are then passed through a PNA block, which will be detailed in later sections, to fuse global information across different scales, resulting in the final five features of different scales.

Based on these features, we then use a robust proposal generator to generate a series of proposals, which will also be detailed in later sections. The proposals generated by this module are then fed into the cascade heads, which consist of three heads that use different thresholds for bounding boxes regression and filtering, to obtain the final results.

### 3.1. Probabilistic Two-Stage Detector Framework

Our probabilistic interpretable framework draws inspiration from CenterNet2 [3]. The aim of an object detector is using bounding boxes to locate objects and provide the class-specific likelihood score for them. Different detectors have similar methods for regressing the bounding boxes, and there is no fundamental difference among them. The core difference lies in how they handle the class likelihood.

One-stage object detectors directly predict the location of the object and its class likelihood. Let $L_{i,c}=1$ represent the *i*th candidate object belongs to the *c*th class(c∈C∪{bg}, C represents the set of all annotated objects; bg means the background class). Although different single-stage object detectors may have different definitions of object and background classes, their overall logic is the same. They maximize the likelihood $P\left(L_{i,c}\right)$ during training and use the class probability to score boxes during inference. One-stage object detectors are a simple, clear, and probabilistically complete framework for object detection.

Two-stage object detectors try to explore as many potential regions of the object as possible in the first stage, and then extract features of these regions again in the second stage and determine their category. Let $O_{i}=1$ present the *i*th potential object location which contains an object; $C_{i}=c$ means it belongs to the *c*th class(c∈C∪{bg}). The goal of the first stage is to maximize the recall of positions with $O_{i}=1$, The goal of the second stage is to maxmize the likelihood $P\left(C_{i}=c|O_{i}=1\right)$. During training, the two stages have different criteria for defining positive samples. The standard in the first stage is loose while the standard in the second stage is strict. During inference, it uses the classification scores of the second stage only. There is no reasonable probability interpretation for the overall detector, for their two stages are disjointed and the training and inference stage are inconsistent.

For the two-stage object detector, a reasonable probability distribution should be Equation (Equation 1):(1)$$\begin{array}{c} P\left(C_{i}=c\right)=P\left(C_{i}=c^{+}|O_{i}=1\right)P\left(O_{i}=1\right)+P\left(C_{i}=bg|O_{i}=1\right)P\left(O_{i}=1\right)\\ +P\left(C_{i}=c^{+}|O_{i}=0\right)P\left(O_{i}=0\right)+P\left(C_{i}=bg|O_{i}=0\right)P\left(O_{i}=0\right) \end{array}$$
where $c^{+}\in C$. It is obvious that the places where $O_{i}=0$ are always lead to the background category. Therefore, the above formula can be further simplified as Equation (Equation 2):(2)$$\begin{array}{cc} P\left(C_{i}=c\right) & =P\left(C_{i}=c^{+}|O_{i}=1\right)P\left(O_{i}=1\right)\\ \; & +P\left(C_{i}=bg|O_{i}=1\right)P\left(O_{i}=1\right)+P\left(O_{i}=0\right) \end{array}$$

We used maximum likelihood estimation to train our detectors in our framework for annotated objects; our goal is to maximize the log-likelihood like Equation (Equation 3):(3)$$\log\left(P\left(C_{i}=c^{+}\right)\right)=\log\left(P\left(C_{i}=c^{+}|O_{i}=1\right)\right)+\log\left(P\left(O_{i}=1\right)\right)$$

The two terms in the above formula correspond exactly to the first and second stages of the detector, respectively. For the background, the maximum-likelihood goal should be Equation (Equation 4):(4)$$\log\left(P\left(C_{i}=bg\right)\right)=\log\left(P\left(C_{i}=bg|O_{i}=1\right)P\left(O_{i}=1\right)+P\left(O_{i}=0\right)\right)$$

However, this objective involves both stages and it does not factorize. In practical applications, it can cause difficulties in back propagation of gradients. Using Jensen’s inequality as in Equation (Equation 5):(5)$$\log\left(\alpha x_{1}+\left(1-\alpha \right)x_{2}\right)\geq \alpha \log\left(x_{1}\right)+\left(1-\alpha \right)\log\left(x_{2}\right)$$ with $\alpha =P\left(O_{i}=0\right)$, $x_{1}=P\left(bg|O_{i}=1\right)$ and $x_{2}=1$, we can get Equation (Equation 6):(6)$$\log P\left(bg\right)\geq P\left(O_{k}=1\right)\log\left(P\left(bg|O_{k}=1\right)\right)$$

It is a tight bound when $P\left(O_{i}=1\right)\to 0$ or $P\left(bg|O_{i}=1\right)\to 1$, and then we add another tight boundary when $P\left(bg|O_{i}=1\right)\to 0$, like Equation (Equation 7):(7)$$\log P\left(bg\right)\geq \log\left(P\left(O_{k}=0\right)\right)$$

The two boundaries mentioned above will be optimized together, so the actual optimization objective for the background class is Equation (Equation 8):(8)$$P\left(O_{k}=1\right)\log\left(P\left(bg|O_{k}=1\right)\right)+\log\left(P\left(O_{k}=0\right)\right)$$

With Equations (Equation 2) and (Equation 8), our first stage maximum represents the likelihood with positive labels at annotated objects and negative labels for all other locations. The first stage of our detector is only used to predict whether there is an object at location O, while the second stage is used to further distinguish the category to which the object belongs. The difference between our detector and traditional two-stage object detectors is that in the training stage, our definition of positive samples is the same for both stages, achieving true end-to-end training. In the prediction stage, we use the scores from both stages to comprehensively evaluate the boxes. The objectives of the two stages of the detector are both maximum likelihood estimation, which has good consistency and relatively complete probability interpretation.

### 3.2. Feature Pyramid

Our feature fusion section references EfficentDet [51] and makes some improvements. It aggregates features from different levels to enable high-level feature maps to contain geometric features from the bottom level, resulting in higher performance of the detector.

Similar to EfficentDet, our feature pyramid is composed of a single block repeated multiple times. The size of each feature map is half of the size of the previous feature map, and all feature maps have the same number of channels. In this paper, we use two forms of feature pyramid: three-layer and five-layer; the blocks that make up them are shown in Figure 2. For the five-layer feature pyramid, the features of the first three layers are taken from the backbone network, while the features of the last two layers are obtained by downsampling the third-layer feature twice; the blocks in Figure 2 are repeated three times. For the three-layer feature pyramid, all the features are taken from the backbone network, and the blocks in Figure 1 are repeated four times.

In terms of feature fusion, we take a five-layer feature pyramid’s block for example. $F_{i-j-m}$ represents features in the middle of the feature fusion process, and $F_{i-j-f}$ means the feature after feature fusion($F_{i-j-f}$ equals to $F_{i-(j+1)}$). Here, we described some fused features as Equation (Equation 9); there will be a batch normalization module and an activation module after each convolution. All convolutions do not change the size of the feature map, and the number of channels in all feature maps is the same.
(9)$$\begin{array}{cc} F_{7-0-f} & =\mathrm{CA}\left(\mathrm{Conv}\left(F_{7-0}\right)\right)\\ F_{6-0-m} & =\mathrm{CA}\left(\mathrm{Conv}\left(F_{6-0}+\mathrm{Pool}\left(F_{7-0}\right)\right)\right)\\ F_{6-0-f} & =\mathrm{CA}\left(\mathrm{Conv}\left(F_{6-0}+F_{6-0-m}+\mathrm{Pool}\left(F_{5-0-f}\right)\right)\right)\\ F_{5-0-m} & =\mathrm{CA}\left(\mathrm{Conv}\left(F_{5-0}+\mathrm{Pool}\left(F_{6-0-m}\right)\right)\right)\\ \;\ldots & \\ F_{3-0-f} & =\mathrm{CA}\left(\mathrm{Conv}\left(F_{3-0}+\mathrm{Pool}\left(F_{4-0-m}\right)\right)\right) \end{array}$$

As shown in Figure 1, we add a channel attention mechanism module to the feature pyramid, because the importance of the information contained in different feature layers is different. By leveraging the significance of inter-channel maps, we can enhance the feature representation of specific semantics, thereby improving the detector’s ability to accurately predict the category of small objects. The channel attention mechanism used in this paper is shown in Figure 3.

We apply the input to a max pooling layer and an average pooling layer separately, with the pooling operation performed along both the width and height axes, resulting in the extraction of features *X* and *Y*; then, we summed them up. We used convolution layers instead of fully connected layers to embed features, thus reducing the computational cost. After two rounds of convolution, we obtained the feature *W*, which represents the importance of each channel. For regularization, we adopted the method of dividing all elements in *W* by the maximum value of *W* instead of using sigmoid, which also aims to reduce computational complexity. To clarify, channel attention is not applied to every repeated FPN but only appears in specific FPN modules, intending to balance accuracy and time. For the five-layer feature pyramid, this module only appears in the second block. For the three-layer feature pyramid, it appears in the second and fourth blocks.

### 3.3. PNA Module

The pyramid non-local attention (PNA) module is the core module of our method, which effectively utilizes the multi-scale and multi-level features generated by the feature pyramid, and establishes dependencies between different locations based on this.

Firstly, let us revisit the definition of non-local attention block, as shown in Figure 4. The input feature map $X\in R^{c\times h\times w}$ goes through three 1 × 1 convolutional layers $W_{\phi }$, $W_{\theta }$ and $W_{\gamma }$, respectively, to obtain three embeddings, namely, $\phi_{0}$, $\theta_{0}$ and $\gamma_{0}\in R^{c^{*}\times h\times w}$, where $c^{*}$ means the channel number after convolution. Then, the three embeddings will be flattened to get ϕ, θ and γ, whose sizes are $c^{*}\times \left(h\times w\right)$. The similarity matrix $M\in R^{\left(h\times w\right)\times \left(h\times w\right)}$ is calculated as Equation (Equation 10):(10)$$M=\mathrm{Norm}\left(\phi^{T}\times \theta \right)$$

Finally, we can get the output *Y* as Equation (Equation 11):(11)$$Y=\mathrm{Conv}\left(\mathrm{Resize}\left(M\times \gamma^{T}\right)\right)$$
where the convolution operation is to adjust the importance of the non-local operation and and restore the channel of the feature map to *c*.

From a spatial perspective, the essence of the non-local attention mechanism is to establish connections between different pixels and regions, as shown in Figure 5a. The output *Y* before performing convolution and resize operations is denoted as $Y^{*}$; for a single location $y_{i}$ in $Y^{*}$, when we choose sigmoid as the normalization method, its relationship with the input *X* is as Equation (Equation 12), where $x_{i}$ means the *i*th location in the input *X*:(12)$$\begin{array}{cc} y_{i} & =\sum_{j}\left[\frac{e^{W_{\phi }{\left(x_{i}\right)}^{T}W_{\theta }\left(x_{j}\right)}}{\sum_{j}e^{W_{\phi }{\left(x_{i}\right)}^{T}W_{\theta }\left(x_{j}\right)}}W_{\gamma }\left(x_{j}\right)\right]\\ \; & =\frac{1}{\sum_{j}e^{W_{\phi }{\left(x_{i}\right)}^{T}W_{\theta }\left(x_{j}\right)}}\sum_{j}e^{W_{\phi }{\left(x_{i}\right)}^{T}W_{\theta }\left(x_{j}\right)}W_{\gamma }\left(x_{j}\right) \end{array}$$

The response $y_{i}$ can incorporate information from all features. However, images of different scales contain varying types of information. For example, reducing the size of an image can filter out some noise and provide purer information. Although the aforementioned operation is effective in capturing long-range correlations, it only extracts information at a single scale. To break this scale constraint, Mei et al. [14] proposed scale-agnostic attention, as shown in Figure 5b, which computes the affinities between a target feature and regions to capture correlations across scales. Let $Z\in R^{c\times \frac{h}{s}\times \frac{w}{s}}$ be the feature map obtained by down-sampling $X\in R^{c\times h\times w}$ by a factor of *s*. Then, $z_{j}$ can be the region descriptor of $x_{\delta (s)}$, where $x_{\delta (s)}$ means the $s^{2}$ neighborhood centred at index *j* on input *x*. The improved formula is as Equation (Equation 13):(13)$$y_{i}=\frac{1}{\sum_{z\in S}\sum_{j\in z}e^{W_{\phi }{\left(x_{i}\right)}^{T}W_{\theta }\left(z_{j}\right)}}\sum_{z\in S}\sum_{j\in z}e^{W_{\phi }{\left(x_{i}\right)}^{T}W_{\theta }\left(z_{j}\right)}W_{\gamma }\left(z_{j}\right)$$

However, the information that can be obtained only by scaling the image is limited. Inspired by this method, as shown in Figure 5c, we will consider fusing scale-agnostic attention with the feature pyramid to achieve a cross-scale non-local attention mechanism. Compared with scaling operations, a feature pyramid can better fuse neighborhood features, extract more abstract and advanced information, and filter out useless noise. The representation of our method is similar to scale-agnostic attention like Equation (Equation 14) where F represents different feature maps, and $f_{j}$ represents the features corresponding to $x_{\delta (s)}$:(14)$$y_{i}=\frac{1}{\sum_{f\in F}\sum_{j\in f}e^{W_{\phi }{\left(x_{i}\right)}^{T}W_{\theta }\left(f_{j}\right)}}\sum_{f\in F}\sum_{j\in f}e^{W_{\phi }{\left(x_{i}\right)}^{T}W_{\theta }\left(f_{j}\right)}W_{\gamma }\left(f_{j}\right)$$

Our detector will use up to five layers of the feature pyramid at most due to the high computational cost of the non-local attention mechanism,; if we directly calculate each point in each feature map, it will cause great computational cost. Looking back at the process of the non-local attention mechanism, we can see that Equations (Equation 10) and (Equation 11) are the main causes of high computational cost, as both equations involve the multiplication of two large matrices. The changes in matrix sizes are as Equation (Equation 15):(15)$$\underset{\phi^{T}}{\underset{︸}{R^{\left(h\times w\right)\times c^{*}}}}\times \underset{\theta }{\underset{︸}{R^{c^{*}\times \left(h\times w\right)}}}\to \underset{M}{\underset{︸}{R^{\left(h\times w\right)\times \left(h\times w\right)}}}\times \underset{\gamma^{T}}{\underset{︸}{R^{\left(h\times w\right)\times c^{*}}}}\to \underset{Y^{*}}{\underset{︸}{R^{\left(h\times w\right)\times c^{*}}}}$$

It can be noticed that the red-highlighted parts do not affect the size of the output $Y^{*}$; therefore, if we adopt some methods to compress the dimensions of the highlighted parts, the computational cost can be greatly reduced.

In our method, we use spatial pyramid pooling (SPP) [61] module, as shown in Figure 6, to compress the dimensions of the highlighted parts. For the non-local attention mechanism on a single feature layer, we first pass $\theta_{0}$ and $\gamma_{0}$ through four pooling layers, to obtain four feature maps of different sizes (1∗1, 3∗3, 6∗6 and 8∗8). Thenm we flatten and concatenate them to obtain $\theta \in R^{c^{*}\times s}$ and $\gamma \in R^{c^{*}\times s}$, where s<<h×w. This can greatly reduce the computational cost. Of course, this does not affect the computational effect, because it is essentially the same as scale-agnostic attention; only the value of *s* in the *s* neighborhood has changed.

The structure of the entire PNA module is shown in Figure 7. The feature maps in the middle layer ($F_{4-3}∼F_{6-3}$) will be fused with the adjacent two layers, while the features in the top layer will only be fused with the previous layer (such as $F_{7-3}$ is only fused with $F_{6-3}$). For the bottom layer feature, such as $F_{3-3}$, it will first be upsampled once through bilinear interpolation to obtain $F_{3-3-up}$, and then undergo subsequent feature fusion. Take feature map $F_{5-3}$ as an example; it will enter the PNA module together with the adjacent feature maps $F_{6-3}$ and $F_{4-3}$. These three features will go through $W_{\gamma }$ and $W_{\theta }$, respectively, and obtain $\gamma_{0-F6}$-$\gamma_{0-F4}$, $\theta_{0-F6}$-$\theta_{0-F4}$. Afterward, this series of features will go through the spatial pyramid pooling (SPP) module, respectively, and each feature map will first generate four different scaled pooling results. Then, the pooling results of each image will be concatenated in order to obtain the feature $\gamma_{F6}$-$\gamma_{F4}$, $\theta_{F6}$-$\theta_{F4}$ with size S×c. $\gamma_{F6}$-$\gamma_{F4}$ will be concatenated again to obtain the feature γ with size 3S×c, and the same applies to θ. The calculation method for feature ϕ is the same as the conventional non-local attention mechanism calculation method. $F_{5-3}$ first goes through a 1×1 convolutional layer $W_{\phi }$, and is then flattened to obtain ϕ. Obviously, the change in the shape of the M matrix does not affect the shape of the final result, although our single PNA module involves three scales at the same time, and the value of 3S is still far smaller than h×w. If the SPP module is not used, our computational complexity will double.

### 3.4. Proposal Generator

The proposal generator in this paper integrates the advantages of various excellent algorithms. The structure of our proposal generator is shown in Figure 1, where the generated feature maps at five scales are fed into the heatmap branch and bbox distribution branch, similar to the GFL [4] algorithm. Considering the issue of blurry boundaries, we generate the distribution of the components related to the box and obtain the final box from the distribution. However, we do not directly generate the four quantities of $\left\{r,l,t,b,\right\}$, but generate them based on the prior anchor boxes, making the network more stable. Subsequently, we encode the distribution of the box and couple it with the heatmap branch to correct the heatmap score. The difference between our proposal generator and the traditional RPN is that we generate fewer but higher-quality proposals and the generated proposals have scores, which plays a role in both training and prediction.

Firstly, for the generation of prior anchor boxes, we conduct k-means clustering on the bounding boxes in the training set to automatically find good priors instead of choosing priors by hand, which is similar to the YOLO [27] series. We adopt the IOU between the prior anchor boxes and the ground truth boxes as the distance metric for clustering to eliminate the influence of box sizes on the error, as in (Equation 16). Finally, we assign the automatically generated anchor boxes to different feature pyramids, with higher levels corresponding to larger proposals.
(16)d(box,centroid)=1−IOU(box,centroid)

Regarding the allocation of ground truth boxes, we use adaptive training sample selection [5]. At each level of the feature pyramid, we choose *k* boxes whose centers are closest to the center of ground truth box gt as the candidate positive samples. After determining the candidate positive samples, we calculate their IOU with the corresponding ground truth boxes and denote the set of all IOU values as $D_{gt}$. We calculate the mean and variance of $D_{gt}$, denoted as $m_{gt}$ and $v_{gt}$, respectively. The threshold value for IOU is set as $t_{gt}=m_{gt}+v_{gt}$. The prior anchor boxes with IOU values greater than or equal to $t_{gt}$ with the ground truth boxes are considered positive samples, as shown in Figure 8. If a prior anchor box satisfies the condition with the IOU values of multiple ground truth boxes, it is assigned to the ground truth box with the highest IOU value.

In complex scenes, the mutual occlusion of objects and blurriness of the main image can lead to uncertainty in the borders, as shown in Figure 9. In this paper, we regress the distribution of the four offset values Δx, Δy, Δh, and Δw based on the borders, and their joint distribution can reflect the clarity of the boundaries. For example, in Figure 9a, when all borders are very clear, the joint distribution of Δx and Δw, and the joint distribution of Δy and Δw, will both have a sharp peak. When one of the upper and lower borders becomes blurry, as in Figure 9b, the peak value of the joint distribution of Δx and Δw will no longer be obvious, and the same goes for the left and right borders. In Figure 9c,d, when the target shows two possible borders, the joint distribution will have two relatively indistinct peaks.

We denote the distribution we predict as F(x), where F(x) satisfies $\int_{-\infty }^{+\infty }F\left(x\right)dx=1$. Let the ground truth be y, and the predicted value by $\hat{y}=\int_{-\infty }^{+\infty }F\left(x\right)x\mathrm{dx}$. We cannot perform calculations and regression on x in the continuous domain, so we artificially add upper and lower boundaries $\left[y_{0},y_{n}\right]$ to x and discretize x to $y_{0},y_{1},y_{2},\ldots ,y_{n}$ to ensure consistency with the convolutional neural network and artificially add upper and lower boundaries, as shown in Equation (Equation 17); in practical algorithms, we use the softmax function as F(x).
(17)$$\hat{y}=\int_{-\infty }^{+\infty }F\left(x\right)x\mathrm{dx}=\sum_{x=y_{0}}^{y_{n}}F\left(x\right)x$$

During training, we want $\hat{y}$ to converge to a value close to y as soon as possible, but we cannot directly calculate the loss between $\hat{y}$ and y; otherwise, regressing $\hat{y}$ through the distribution will lose its meaning. The value of the ground truth y is not necessarily exactly one of $y_{0}-y_{n}$. Therefore, in this case, we choose to make the distribution as close as possible to two adjacent values $y_{i}$ and $y_{i+1}$ of y. Taking the joint distribution of Δx and Δw as an example, assuming the ground truth is obtained at $\Delta x^{*}$ and $\Delta w_{*}$, we want the joint distribution of Δx and Δw to converge to $\Delta x^{i}$$\Delta x^{i+1}$ and $\Delta w^{i}$$\Delta w^{i+1}$ as soon as possible. The design of the loss function is as Equation (Equation 18):(18)$$\begin{array}{cc} \mathrm{Loss}(\Delta x,\Delta x^{*},\; & \Delta w,\Delta w^{*}=\\ \; & -\left(\left(\Delta x_{i+1}-\Delta x^{*}\right)\log\left(F\left(\Delta x_{i+1}\right)\right)+\left(\Delta x^{*}-\Delta x_{i}\right)\log\left(F\left(\Delta x_{i}\right)\right)\right)\\ \; & -\left(\left(\Delta w_{i+1}-\Delta w^{*}\right)\log\left(F\left(\Delta w_{i+1}\right)\right)+\left(\Delta w^{*}-\Delta w_{i}\right)\log\left(F\left(\Delta w_{i}\right)\right)\right) \end{array}$$
where:(19)$$F\left(\delta x_{i}\right)=\frac{e^{\Delta x^{i}}}{\sum_{j=0}^{n}e^{\Delta x^{j}}},F\left(\delta x_{i+1}\right)=\frac{e^{\Delta x^{i+1}}}{\sum_{j=0}^{n}e^{\Delta x^{j}}}$$

For the heatmap branch, we use soft one-hot encoding to label the ground truth, which is different from the traditional method where the value of positive sample points is all 1 and the value of negative sample points is all 0. We assign a value of 0 < y ≤ 1 to the positive sample points, where y is the IOU score of the point, and the larger the IOU value between the anchor and the ground truth at the point, the larger the value of y. The advantage of this approach is that it establishes a connection between the position and the IOU, making the consistency of the network better during training and prediction. At the same time, positive samples with a higher ground truth IOU can contribute more weight, thereby improving the performance of the network.

In the follow-up process, we will encode the distribution of the border distribution branch and apply the result to the heatmap. The specific process is shown in Figure 1. First, we select the top k values from the discrete distribution and then input them into two FC layers and an activation layer to generate corresponding weights, which are multiplied by the corresponding points on the heatmap. The reason is that the distribution of bounding boxes is strongly correlated with the IOU score. Coupling the two branches can further improve the accuracy of the heatmap and reduce the difficulty of training, making the proposal score more accurate.

### 3.5. Cascade Heads

In this paper, we adopt cascade heads as the second stage of our detector, which decompose the regression of categories and bounding boxes into multiple stages; each stage takes the bounding boxes from the previous stage along with the feature map as inputs, and outputs the classification and a new distribution of bounding boxes. The detailed structure of cascade heads is illustrated in Figure 10.

Regarding the bounding box regression part, it relies on a cascade of specialized regressors, as depicted in Equation (Equation 20).
(20)$$f\left(x,b\right)=f_{T}\circ f_{T-1}\circ \cdots \circ f_{1}\left(x,b\right)$$

In this formula, *x* represents the input feature map, and *T* represents the total number of stages. In this paper, we set T=3. Each stage has an independent regressor $f_{t}$ with independent parameters, instead of simply repeating the same *f* multiple times. The cascaded regression is a resampling procedure that changes the distribution of hypotheses to be processed by the different stages. Likewise, each regressor *f* in the cascade is optimized based on the sample distribution $\left\{b^{t}\right\}$ that arrives at the corresponding stage, rather than the initial distribution of $\left\{b^{0}\right\}$. The cascade progressively enhances hypotheses. The cascade heads utilize the same structure and parameters during both training and inference; this provides a more reasonable probability explanation and there is no discrepancy between training and inference distributions.

As the number of regressions increases, the quality of the bounding boxes improves; in other words, the cascade regression begins with a set of examples $\left\{b^{i}\right\}$, and then iteratively samples a new example distribution $\left\{b^{i+n}\right\}$ with a higher IoU. Therefore, to maintain a relatively balanced number of positive samples and to maximize the elimination of outliers in order to enable a better trained sequence of specialized detectors, the regressors in different stages should use different IOU thresholds, and the IOU thresholds should be increased gradually. In practical training, our three regressors use $\left\{0.5,0.6,0.7\right\}$ as IOU thresholds, which is consistent with the original paper.

As for the classification part, each cascade head has an independent classification branch with different parameters, which outputs the probability of the target belonging to each class. Unlike the bounding box regression part, the classification results of each stage are not affected by the results of the previous stage. The cascade heads is learned by minimizing the loss in Equation (Equation 21). where $b^{t}=f_{t-1}\left(x^{t-1},b^{t-1}\right)$, *g* is the ground truth, $h_{t}$ is the classifier of the t-th cascade head.
(21)$$L\left(x^{t},g\right)=L_{cls}\left(h_{t}\left(x^{t}\right),y^{t}\right)+\lambda \left[y^{t}\geq 1\right]L_{loc}\left(f_{t}\left(x^{t},b^{t}\right),g\right)$$

During the prediction phase, we also couple the two stages. Specifically, the score of the final bounding box is obtained by multiplying the score of the first stage with the score of each cascade. This is one of the essential differences between our method and traditional two-stage object detectors, as the two stages of our detector are not separate.

## 4. Results

To demonstrate the effectiveness of our algorithm, we conducted comparisons with baselines and ablation experiments on the COCO dataset, including object detection and instance segmentation tasks, and provided detailed explanations for the performance of each part of our algorithm. We also compared our algorithm with state-of-the-art algorithms on both the COCO [62] and Pascal VOC [63] datasets, achieving the best performance when using the same backbone. Finally, we further tested our algorithm on domestic care robots and four-wheel unmanned platforms, compared with baselines, and demonstrated the superiority of our algorithm in practical application scenarios.

### 4.1. Ablation Study

The architecture of our method is inspired by CenterNet2, so we used CenterNet2 as the baseline for comparison. As mentioned earlier, the core difference in the structure between our method and CenterNet2 as well as other two-stage object detectors lies in the generation and use of proposals in the first stage. Our algorithm can be seen as CenterNet2 with a replacement of the proposal generator, the addition of the PNA module, and modifying part of the FPN. This experiment was conducted on the COCO dataset, and all methods used DLA as the backbone and a three-layer FPN for feature fusion. All methods were trained for 60 epochs, using 0.02 as the base learning rate and 640*640 as the base training size. No data augmentation methods were used except for random cropping and random resizing. The training and evaluation were performed on Intel Xeon6130 processor and a single TITANxp GPU with PyTorch 1.10.0 and CUDA 10.2. The ablation experiment results for object detection tasks are shown in Table 1.

Comparing the results of object detection, it can be seen that our algorithm has a significant advantage in accuracy compared to CenterNet2. Our method can better detect small targets and capture details. Further analysis of the table shows that the PNA module contributes the most to the algorithm’s performance, followed by our robust proposal generator. Although the channel attention mechanism module we designed has the smallest contribution to the overall accuracy improvement, it hardly affects the efficiency of the algorithm.

To further explore the principles of our various modules, we conducted the following work. As mentioned earlier, the reason why the PNA module can significantly improve the algorithm’s performance is that it can establish feature connections between long distances and different feature layers, allowing the network to better focus on important information and restore details. We separated the feature maps output by the feature pyramid during the prediction process, selected the k channels with the highest activation in the feature maps, generated a heatmap, and superimposed it on the original image, as shown in Figure 11. The deeper the red color, the higher the value of the heatmap, indicating that the region has a higher activation and is more focused by the network. It can be seen that our algorithm pays more attention to small targets; small targets that CenterNet did not focus on are also well attended to after adding the PNA module. Additionally, when there are many targets in the scene, our attention is more concentrated and the activation intensity is higher. This also demonstrates the role of channel attention mechanism, which allows channels with higher activation to have higher weights and perform better in subsequent tasks.

The detection performance of our algorithm and its comparison with CenterNet2 are shown in Figure 12. Thanks to the application of PNA, our algorithm has better performance on small targets, occluded targets, blurry targets, and hidden targets in complex backgrounds.

The advantage of our robust proposal generator compared to RPN and other proposal generators is that it can generate higher-quality proposals. Specifically, it generates fewer proposals, but with a higher IOU with the ground truth, as shown in Figure 13. We compared the performance of our method with the proposal generators in traditional RPN and CenterNet2. The advantages of our algorithm become more evident when there are more items in the scene and they are arranged in a more disorderly manner. The reason for the above results is that the use of prior boxes can to some extent avoid the generation of proposals that are too large or too small. Additionally, coupling the box distribution branch with the heatmap branch and using soft one-hot encoding associated with IOU can make proposals with higher IOU with the ground truth have higher scores and be more easily retained, while poor quality boxes with small IOU with the ground truth are more easily eliminated.

In addition, thanks to the coupling of the bounding box distribution branch and the heatmap generation branch, we fully utilized the distribution information of the bounding box. It can be seen that when multiple targets overlap and the target bounding box is blurred, our false positive rate is significantly lower, and the bounding boxes we regress are more reasonable, as shown in Figure 14.

After adding a simple segmentation head, the method we constructed can complete the instance segmentation task. Further comparison with CenterNet2 was conducted with the same segmentation head on the COCO dataset instance segmentation task under the same training environment. The experimental results are shown in Table 2 and Figure 15. As can be seen, our method has more accurate boundary segmentation, especially in complex scenarios.

### 4.2. Experiment on COCO Dataset

Table 3 compares our algorithm with some existing advanced algorithms. To better explore the performance of our algorithm, we used data augmentation methods such as random cropping, blur, and random contrast, and used cosine annealing learning rate decay, the base training size is still 640*640. We trained and predicted on Intel 127,00K processor and two Nvidia RTX3090 GPUs with PyTorch 1.10.0 and CUDA 11.3. It can be seen that when using the same backbone, our algorithm performs better than some current algorithms. When using ResNXet-101 as the backbone, our algorithm can achieve an accuracy of 51.3. In addition, compared to CenterNet2, we always have better results when using the same backbone, and our advantages are particularly evident in small objects.

In addition to conducting experiments on object detection tasks on the COCO dataset, we also conducted experiments on instance segmentation tasks. The training strategy and environment were the same as those for object detection tasks. Although we only added a simple segmentation head based on the original algorithm, our algorithm still performed better than current mainstream algorithms when using the same backbone. The comparison results are shown in Table 4.

### 4.3. Experiment on Pascal VOC dataset

Table 5 reports object detection results on the PascalVOC dataset; the training environment, strategy, and related hyperparameters are the same as those in the COCO experiment. We train on VOC 2007 and VOC 2012 trainval sets and test on VOC 2007 test set. We can achieve a high AP value on this dataset and have certain advantages compared to current mainstream algorithms.

### 4.4. Experiment on Our Platform

#### 4.4.1. Household Serving Robot

Our first robot platform, as shown in Figure 16, is a household serving robot that has the functions of autonomous recognition, picking up and delivering corresponding items according to instructions, and operating home appliances. Its workflow is roughly shown in Figure 17a, and our algorithm is a key part of the process, providing the location of the target to be grabbed for the robot.

We built our own dataset by combining the actual working scenarios of the robot with the target to be grabbed. The dataset consists of 2300 images, 3443 instances totally, with 1800 images for training and 500 images for testing. The performance of our method on the dataset is shown in Table 6; the evaluation criteria are consistent with COCO.

The application of our algorithm in actual scenarios is shown in Figure 18. Our algorithm can handle various scenarios, including blurred images caused by the robot’s rapid movement, poor indoor lighting conditions, and scenes where multiple targets overlap with each other.

#### 4.4.2. Rebar-Binding Robot

Our second robot platform, as shown in Figure 19, is an autonomous rebar-binding robot for construction. It has the functions of recognizing rebar intersection points, binding rebar, and determining whether the tied rebar at the intersection point was bound (as shown in Figure 20). Its workflow is roughly shown in Figure 17b. Our algorithm is used to detect the intersection points and determine whether the intersection points are tied properly.

Similarly, we have also built a dataset of rebar intersection points, with 260 images used for training and 50 images used for testing, 9150 totalling instances. As shown in Figure 20, the left side of the picture is the intersection point that has been bound, which is recorded as 0 in the data set, and the right side is the intersection point of the steel bar that has not been bound, which is recorded as 1. Because the use scenario of our robot is construction sites, the algorithm is affected not only by complex lighting conditions but also by ground cracks and steel reflections. Therefore, it is a challenging task. The performance of our method on the dataset is shown in Table 7.

The performance of our algorithm in practical scenarios is shown in Figure 21. It can be seen that our algorithm has a very high detection success rate and a very low false detection rate, and it performs very well even under extreme dark lighting conditions and serious interference from steel reflections and ground cracks.

## 5. Conclusions

In this study, we proposed a probabilistic two-stage object detector. The detector has a relatively compact structure and a better probability interpretation, which leads to higher accuracy, stronger adaptability, and greater sensitivity to small objects. We proposed a strong proposal generator as the first stage of the detector. The generator uses a more reasonable ground-truth matching method and takes into account the case of blurred object boundaries. Its bounding box distribution branch is coupled with the heatmap branch, allowing the generator to make full use of various information. Our generator can generate proposals with scores that have higher IOU with ground truth. Furthermore, we proposed the PNA module, which combines the non-local attention mechanism with the feature pyramid. This module breaks the limitation of scale for non-local attention mechanisms and greatly enhances the detector’s ability to mine details and comprehend global semantic information. We also integrated the SPP module into the non-local attention mechanism to reduce computational costs.

Subsequent experiments have demonstrated the superiority of our method. Our method achieved outstanding performance in both detection and segmentation tasks on the COCO dataset and outperformed most mainstream algorithms on the Pascal VOC dataset. Moreover, we applied our method to challenging scenarios in construction sites and demonstrated its excellent performance in completing various tasks. However, our algorithm still has certain limitations. Future research can explore how to compress the algorithm’s time to achieve more efficient object detection.

## Figures and Tables

**Figure 1 sensors-23-04938-f001:**
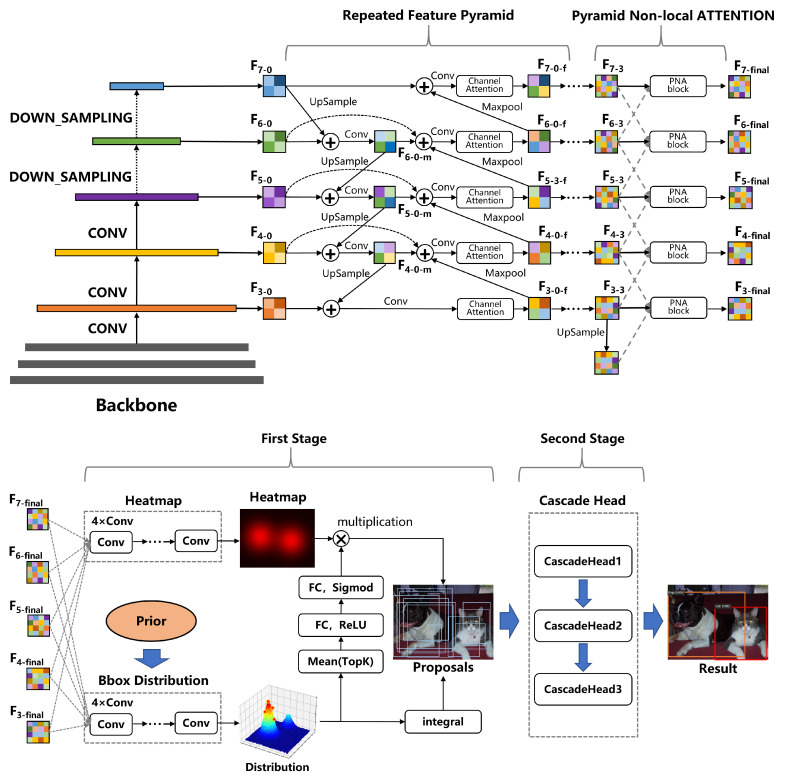
The architecture of our proposed probabilistic two-stage object detector.

**Figure 2 sensors-23-04938-f002:**
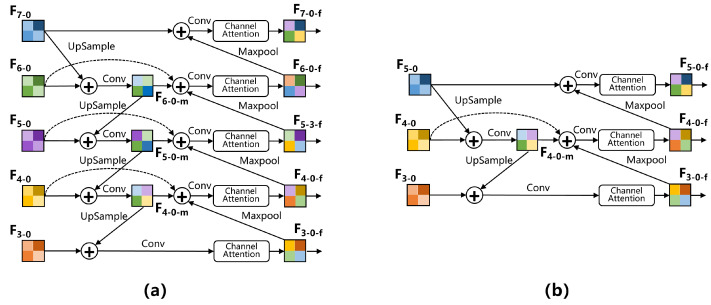
The architecture of (**a**) the single block of the five-layer feature pyramid, (**b**) the single block of the three-layer feature pyramid.

**Figure 3 sensors-23-04938-f003:**
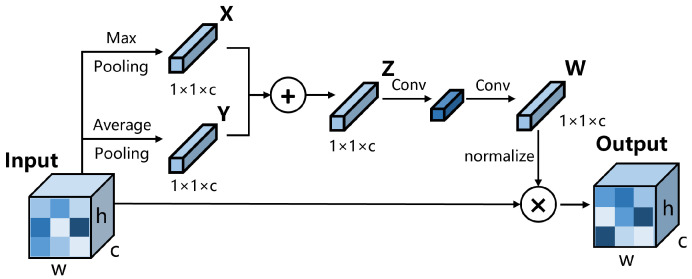
The architecture of our channel attention module.

**Figure 4 sensors-23-04938-f004:**
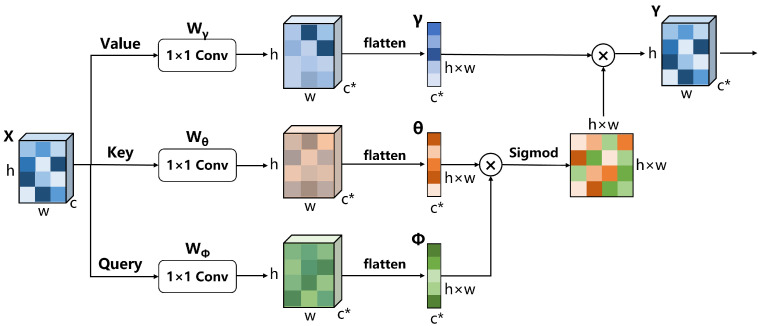
A schematic diagram of non-local attention.

**Figure 5 sensors-23-04938-f005:**
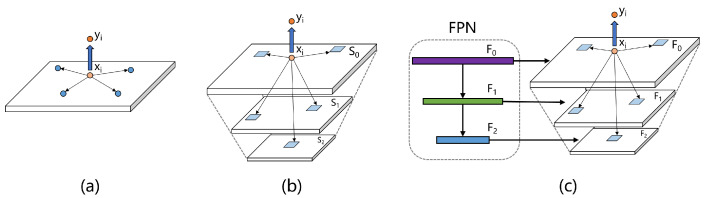
A schematic diagram of (**a**) non-local attention, (**b**) scale-agnostic non-local attention, (**c**) our pyramid attention.

**Figure 6 sensors-23-04938-f006:**
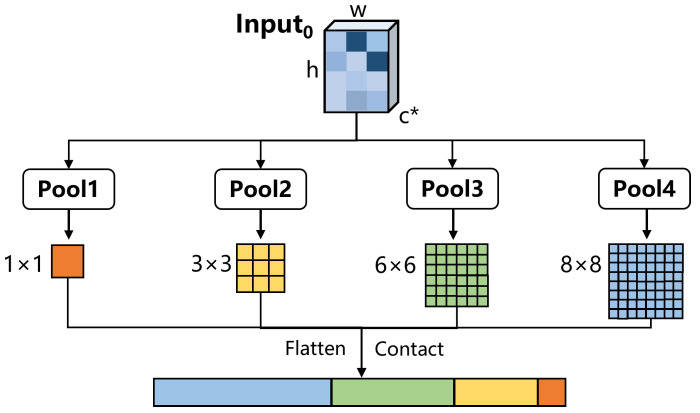
The architecture of spatial pyramid pooling module.

**Figure 7 sensors-23-04938-f007:**
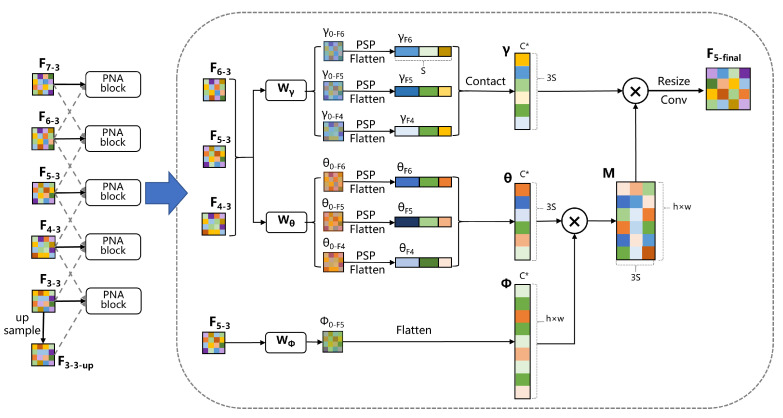
The architecture of our pyramid non-local attention (PNA) module.

**Figure 8 sensors-23-04938-f008:**
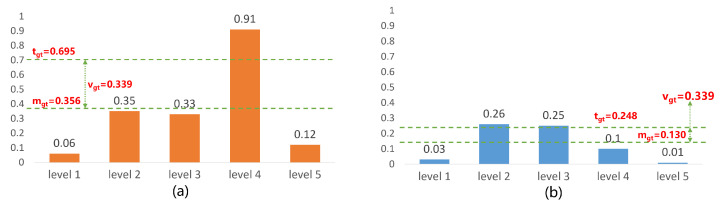
Illustration of sample selection, suppose there is only one candidate box per level. (**a**) A gt with a high mg and a high vg, the candidate box from level 4 will be chosen, (**b**) A gt with a low mg and a high vg, the candidate boxes from level 2 and level 3 will be chosen.

**Figure 9 sensors-23-04938-f009:**
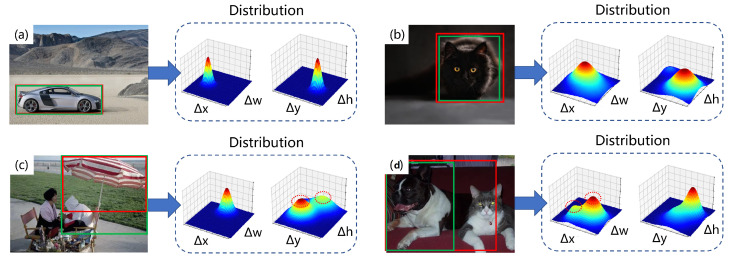
The joint distribution of Δx and Δw, and the joint distribution of Δy and Δh under different circumstances.

**Figure 10 sensors-23-04938-f010:**
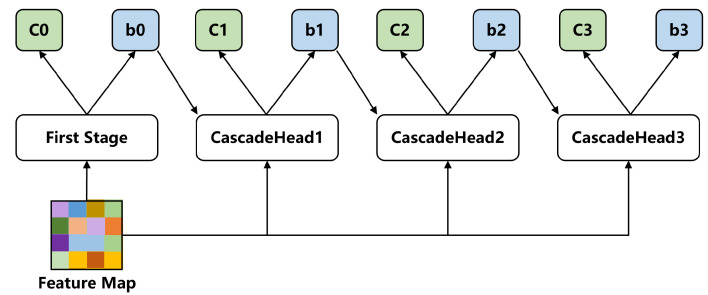
A schematic diagram of non-local attention.

**Figure 11 sensors-23-04938-f011:**
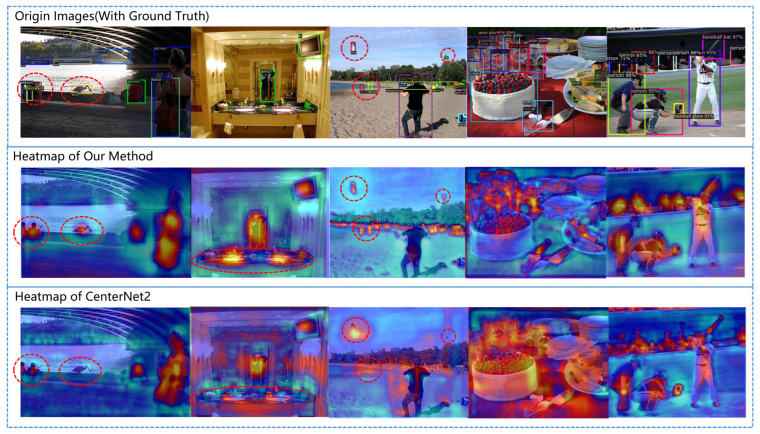
Comparison between our heat maps and CenterNet2’s heat maps.

**Figure 12 sensors-23-04938-f012:**
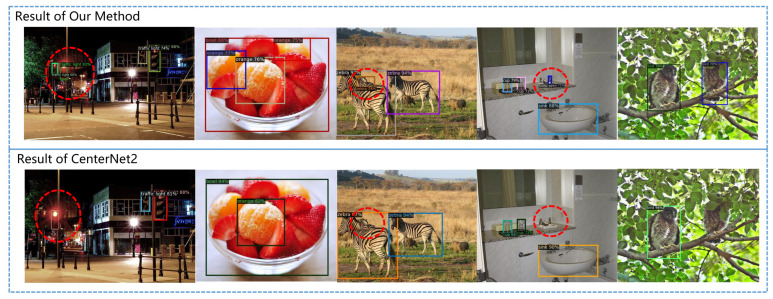
Comparison of detection results between our method and CenterNet2 for small objects, occluded objects, and partially hidden objects.

**Figure 13 sensors-23-04938-f013:**
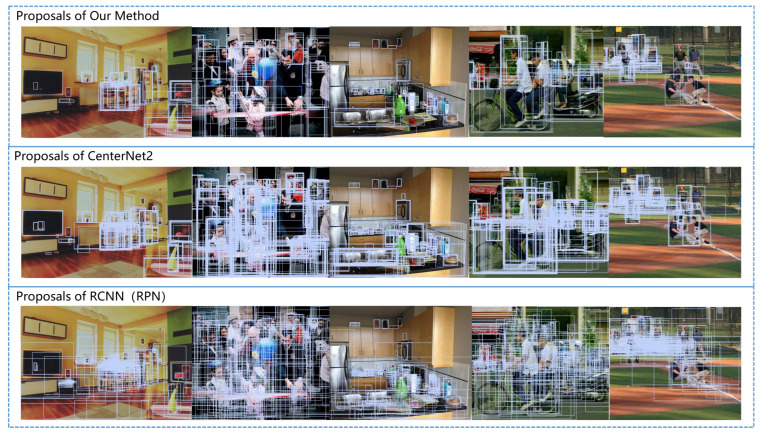
Comparison between the proposals generated in the first stage of our method, CenterNet2, and traditional RPN. For clarity, we only show regions with score >0.3.

**Figure 14 sensors-23-04938-f014:**
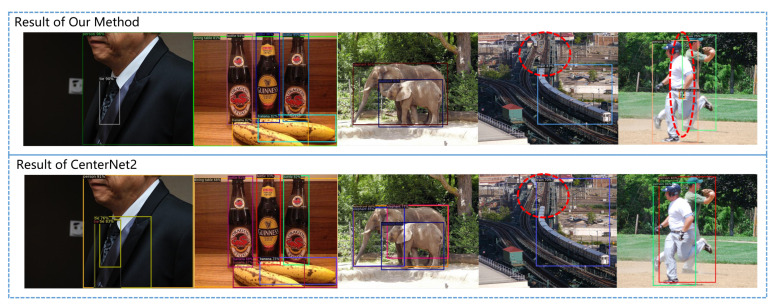
Comparison of detection results between our method and CenterNet2 in scenarios with multiple overlapping objects or blurry object boundaries.

**Figure 15 sensors-23-04938-f015:**
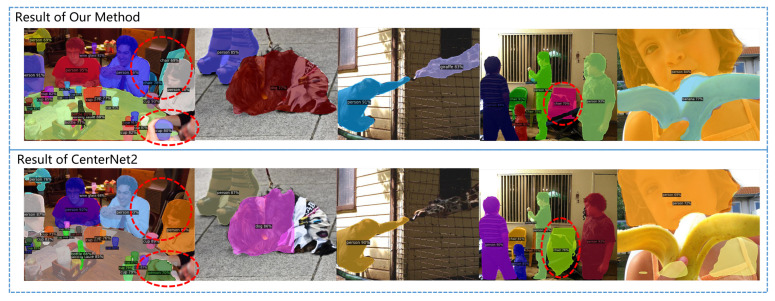
Comparison between our method and CenterNet2 on instance segmentation task of COCO dataset.

**Figure 16 sensors-23-04938-f016:**
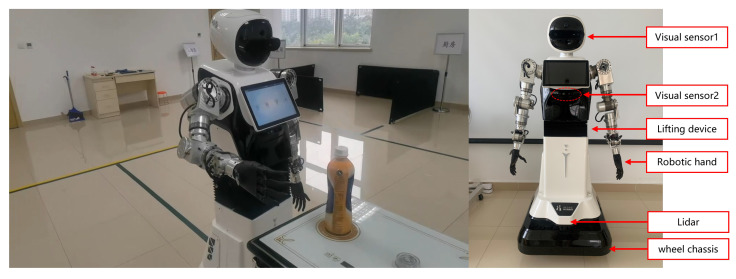
The photo of our household serving robot platform.

**Figure 17 sensors-23-04938-f017:**
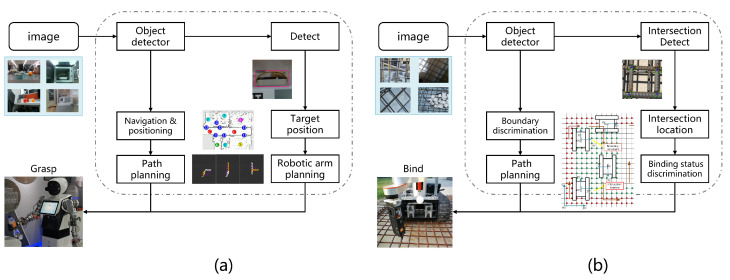
The workflow of household serving robot and rebar binding robot.

**Figure 18 sensors-23-04938-f018:**
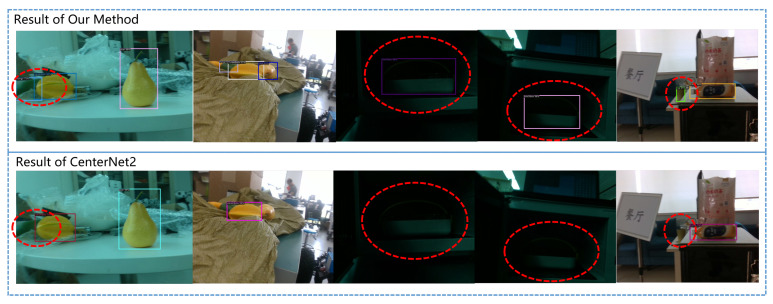
The comparison of our method with CenterNet2 in practical application scenarios.

**Figure 19 sensors-23-04938-f019:**
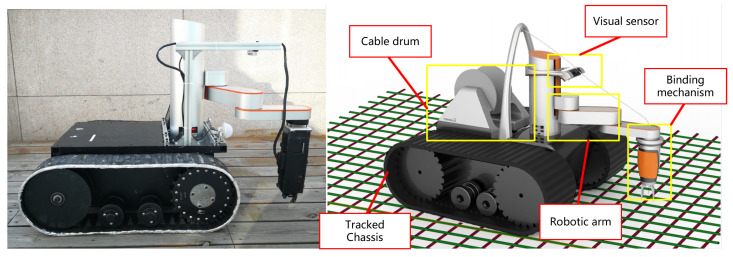
The photo of our rebar-binding robot platform.

**Figure 20 sensors-23-04938-f020:**
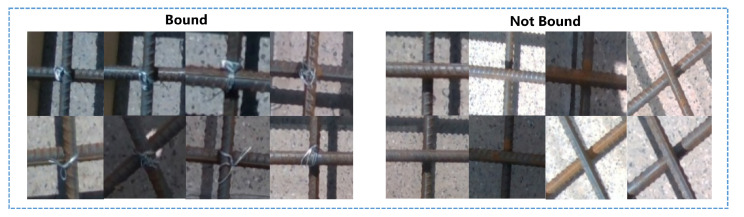
Some pictures in our rebar data set.

**Figure 21 sensors-23-04938-f021:**
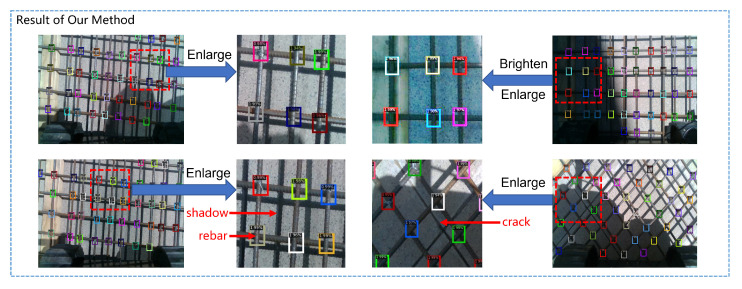
The performance of our algorithm in practical scenarios.

**Table 1 sensors-23-04938-t001:** Ablation experiments on object detection task of COCO dataset.

Method ^1^	Run Time	AP	${\mathrm{AP}}_{50}$	${\mathrm{AP}}_{75}$	${\mathrm{AP}}_{S}$	${\mathrm{AP}}_{M}$	${\mathrm{AP}}_{L}$
CenterNet2-p3	36 ms	43.7	60.3	47.5	23.5	48.1	59.5
CenterNet2-p3*	37 ms	43.9(+0.2)	60.3	47.8	23.6	48.4	59.6
CenterNet2-p3+pg	39 ms	44.5(+0.8)	61.4	48.7	24.9	48.5	59.9
CenterNet2-p3+PNA	54 ms	46.4(+2.7)	64.0	50.4	27.5	51.4	62.0
CenterNet2-p3*+pg+PNA (ours)	58 ms	**47.0(+3.3)**	**64.4**	**51.1**	**27.9**	**51.9**	**62.6**

^1^ “CenterNet2-p3” represents CenterNet2 using the original proposal generator and the FPN structure from the EfficientNet paper. “CenterNet2-p3*” represents CenterNet2 using the original proposal generator and an improved FPN structure. “+pg” indicates that our proposal generator was used instead of the original one in the CenterNet2-p3* model.

**Table 2 sensors-23-04938-t002:** Results of our method and CenterNet2 on instance segmentation task of COCO dataset.

Method	Run Time	AP	${\mathrm{AP}}_{50}$	${\mathrm{AP}}_{75}$	${\mathrm{AP}}_{S}$	${\mathrm{AP}}_{M}$	${\mathrm{AP}}_{L}$
CenterNet2	41 ms	33.8	55.9	33.0	14.7	36.5	51.0
PNANet (our method)	66 ms	**35.2**	**57.3**	**36.5**	**15.7**	**37.9**	**53.8**

**Table 3 sensors-23-04938-t003:** Performance of state-of-art methods and our method on object detection tasks of COCO dataset.

Method	Backbone ^1^	AP	${\mathrm{AP}}_{50}$	${\mathrm{AP}}_{75}$	${\mathrm{AP}}_{S}$	${\mathrm{AP}}_{M}$	${\mathrm{AP}}_{L}$
CenterNet [33]	DLA34	41.6	60.3	45.1	21.5	43.9	56.0
CenterNet2-p3 [34]	DLA34	43.7	60.3	47.5	23.5	48.1	59.5
PNANet-p3(ours)	DLA34	**47.0**	**64.4**	**51.1**	**27.9**	**51.9**	**62.6**
RefineDet [64]	R101	41.8	62.9	45.7	25.6	45.1	54.1
Cascade RCNN [18]	R101	42.8	62.1	46.3	23.7	45.5	55.2
ATSS [5]	R101	43.6	62.1	47.4	26.1	47.0	53.6
Conditional DETR [65]	R101	44.5	65.5	47.5	23.6	48.4	63.6
PAA [34]	R101	44.8	63.3	48.7	26.5	48.8	56.3
GFLV2 [66]	R101	46.2	64.3	50.5	27.8	49.9	57.0
CenterNet2-p5 [3]	R101	43.5	59.8	48.2	24.2	47.9	59.2
PNANet-p5(ours)	R101	**46.6**	**63.6**	**50.5**	**27.0**	**51.4**	**62.0**
Cascade RCNN [18]	X101	48.8	67.7	52.9	29.7	51.8	61.8
ATSS [5]	X101	47.7	66.6	52.1	29.3	50.8	59.7
Deformable DETR [42]	X101	50.1	69.7	54.6	30.6	52.8	65.6
PAA [34]	X101	49.0	67.8	53.3	30.2	52.8	62.2
GFL [4]	X101	48.2	67.4	52.6	29.2	51.7	60.2
AutoAssign [67]	X101	49.5	68.7	54.0	29.9	52.6	62.0
CenterNet+ [33]	X101	49.1	67.8	53.3	30.2	52.4	62.0
CenterNet2 [3]	X101	50.2	68.0	55.0	31.2	53.5	63.6
PNANet-p5(ours)	X101	**51.3**	**69.1**	**55.8**	**34.1**	**55.6**	**65.6**

^1^ “R101” represents ResNet-101, “X101” represents ResNeXt-101.

**Table 4 sensors-23-04938-t004:** Performance of state-of-art methods and our method on instance segmentation task of COCO dataset.

Method	Backbone	AP	${\mathrm{AP}}_{50}$	${\mathrm{AP}}_{75}$	${\mathrm{AP}}_{S}$	${\mathrm{AP}}_{M}$	${\mathrm{AP}}_{L}$
MNC [68]	R101	24.6	44.3	24.8	4.7	25.9	43.6
FCIS [69]	R101	29.2	49.5	29.5	7.1	31.3	40.0
Mask-RCNN [20]	R101	33.1	54.9	34.8	12.1	35.6	51.1
PolarMask [70]	R101	30.4	51.9	31.0	13.4	32.4	42.8
CenterNet2-p5 [3]	R101	33.4	54.2	34.3	15.0	35.5	50.4
PNANet-p5 (ours)	R101	**34.6**	**56.5**	**35.7**	**16.7**	**37.1**	**53.6**

**Table 5 sensors-23-04938-t005:** Performance of mainstream methods and our method on Pascal VOC dataset.

Method	Backbone	AP@50 ^1^
Faster RCNN [17]	R101	79.8
R-FCN [71]	R101	80.5
SSD [72]	R101	78.9
DSSD [73]	R101	81.5
CenterNet [33]	R101	78.7
CenterNet2 [3]	R101	79.6
PNANet(ours)	R101	**81.9**

^1^ The results are shown in mAP@0.5, consistent with the CenterNet paper, rather than VOC-11 points.

**Table 6 sensors-23-04938-t006:** Results of our method and CenterNet2 on our household serving robot dataset.

Method	AP	${\mathrm{AP}}_{50}$	${\mathrm{AP}}_{75}$
CenterNet2	81.9	97.2	94.6
PNANet (our method)	84.7	98.7	96.0

**Table 7 sensors-23-04938-t007:** Results of our method and CenterNet2 on our rebar-binding robot dataset.

Method	AP	${\mathrm{AP}}_{50}$	${\mathrm{AP}}_{75}$
CenterNet2	87.2	98.9	95.9
PNANet (our method)	89.1	99.5	96.4

## Data Availability

The data presented in this study are available on request from the corresponding author. The data are not publicly available due to policy reasons.

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
