# Peer review of "PNANet: Probabilistic Two-Stage Detector Using Pyramid Non-Local Attention"

_sensors, 2023, doi:10.3390/s23104938_

Round 1

Reviewer 1 Report

Please kindly find the attached file.

The quality of the English language of the presented article is proper. If they can apply minor revisions, it will be great.

Reviewer 2 Report

The paper presents a high degree of originality. The math is well presented and the pictures clearly describe the procedures.

Reviewer 3 Report

  1. Please proofread the manuscript for grammar and format errors. For example, lines 16, 87, 148 (top-down pathway structure of what? For what purpose?). Also, please list the full name of the abbreviations when they first appear. For example, GFL, ATSS, FPN, etc.
  2. What are long-range dependencies in lines 57-66? Please consider listing some examples to improve the readability.
  3. In line 130, please consider adding any example numbers to support “they still lack accuracy”. Otherwise, it is hard to agree with this conclusion.
  4. Lines 198-201 said the CascadeHead is not discussed here, while the Figure 1 shows the CasadeHead is the second stage of the proposed model. Since this manuscript emphasizes the two-stage object detector, the CascadeHead should be covered in this manuscript as a comprehensive study.

Minor revisions are needed on the grammar and format.
